# $A^*$ SAMPLING WITH PROBABILITY MATCHING

## ABSTRACT

Probabilistic methods often need to draw samples from a nontrivial distribution. $A^*$ sampling is a nice algorithm by building upon a top-down construction of a Gumbel process, where a large state space is divided into subsets and at each round $A^*$ sampling selects a subset to process. However, the selection rule depends on a bound function, which can be intractable. Moreover, we show that such a selection criterion can be inefficient. This paper aims to improve $A^*$ sampling by addressing these issues. To design a suitable selection rule, we apply *Probability Matching*, a widely used method for decision making, to $A^*$ sampling. We provide insights into the relationship between $A^*$ sampling and probability matching by analyzing a nontrivial special case in which the state space is partitioned into two subsets. We show that in this case probability matching is optimal within a constant gap. Furthermore, as directly applying probability matching to $A^*$ sampling is time consuming, we design an approximate version based on Monte-Carlo estimators. We also present an efficient implementation by leveraging special properties of Gumbel distributions and well-designed balanced trees. Empirical results show that our method saves a significantly amount of computational resources on suboptimal regions compared with $A^*$ sampling.

## 1 INTRODUCTION

Probabilistic methods provide an important family of tools in machine learning for modeling uncertainty of complex systems, performing probabilistic inference, revealing hidden factors (Ghahramani, 2015), and making decisions Kocsis & Szepesvári (2006). These methods usually involve a fundamental task of drawing samples from a nontrivial distribution.

There exists a lot of work approaching the sampling problems, including rejection sampling (Gilks & Wild, 1992), MCMC (Propp & Wilson, 1996), etc. Recently, sampling algorithms based on the Gumbel process have received increasing attentions (Malmberg, 2013; Hazan et al., 2013; Gane et al., 2014; Hazan & Jaakkola, 2012; Papandreou & Yuille, 2011; Tarlow et al., 2012; Kappes et al., 2015; Kim et al., 2016) since a Gumbel process can turn a sampling task to an optimization problem so that we can use optimization tools to finish the original sampling task.

In this work, we focus on $A^*$ sampling (Maddison et al., 2014) which is one of the most famous Gumbel process based sampling algorithm. The major advantage of $A^*$ sampling is that it can be applied to large state spaces, e.g., a continuous sample space or a discrete space whose size is exponentially large. The reason is that $A^*$ sampling divides the state space into disjoint subsets and takes each subset as a whole, so that it can avoid initializing a large number of states, which is often encountered by other Gumbel process based algorithms (Papandreou & Yuille, 2011). Furthermore, $A^*$ sampling adaptively selects subsets to process and the performance of $A^*$ sampling is highly dependent on the selection rule.

However, how to select subsets to process is very challenging. In each round, $A^*$ sampling processes the subset with maximum $D(S)$ which is an upper bound of the maximum Gumbel value within a subset $S$ (see Section 2 for more details of $D(S)$). But in general, it is difficult to compute $D(S)$ since it is an instance of non-convex optimization. Another challenge is that even if we are able to compute $D(S)$ efficiently, selecting a subset with the maximum $D(S)$ may not be a good choice. This is because our target is to process subsets with larger Gumbel values, but $D(S)$ only provides an upper bound. So it is possible that the Gumbel value of $S$ is relatively small with high probability while $D(S)$ is very large. In this case, $A^*$ sampling will waste many computational resources on

suboptimal regions. We'll discuss more on how this inaccuracy of $D(S)$ deteriorates the performance of $A^*$ sampling by analyzing a counter example in Section 3.

To address the above challenges, we improve the subset selecting procedure of $A^*$ sampling with probability matching (PM) which has been proven efficient in many settings of making decisions, including Bayesian bandits (Chapelle & Li, 2011), MDP (Osband & Van Roy, 2016), economic decisions (Vulkan, 2000), etc.

**Contributions:** Intuitively, PM randomly selects an option according to its probability of being the optimal, so that it won't select a suboptimal option for too many rounds. To provide more insights into the efficiency of applying PM to $A^*$ sampling, we first analyze a simple but nontrivial special case in which the state space is partitioned into two subsets. As we'll present in Section 4.1, in this case, PM is optimal within a constant gap in terms of the stochastic regret (Guha & Munagala, 2014) which measures the number of selected rounds on suboptimal options. Furthermore, as directly applying PM to $A^*$ sampling is time consuming, we design a novel approximate algorithm based on Monte-Carlo estimators. The approximate algorithm is computationally efficient since it utilizes special properties of Gumbel distributions and well-designed balanced trees. We empirically compare our method with popular baselines of $A^*$ sampling and Metropolis-Hastings algorithm. Experiments show that our algorithm works well.

## 2 PRELIMINARIES

In this section, we present some preliminary knowledge of the Gumbel process and $A^*$ sampling. Below, we first introduce basic definitions of probability distributions and Gumbel distributions.

**Definition 1** (Probability distributions). *In general, a distribution $P$ on a state space $\Omega$ provided its potential function, $\phi_P : 2^\Omega \to \mathbb{R}$, is a sigma-finite measure such that $P(S) = \frac{1}{Z_P} \exp\{\phi_P(S)\}$, where $Z_P = \exp(\phi_P(\Omega))$ is normalizing constant.*

**Definition 2** (Gumbel and Truncated Gumbel distributions (Malmberg, 2013)). *Let $c$ denote the Euler constant. For convenience, define $e_\psi(g) = \exp(-g + \psi)$, $F_\psi(g) = \exp(-\exp(-g + \psi))$ and $f_\psi(g) = e_\phi(g)F_\psi(g)$. Then (1), $\mathcal{G}(\psi)$: a Gumbel distribution with location $\psi$ has PDF and CDF at state $g$: $f_{\psi+c}(g), F_{\psi+c}(g)$. (2), $\mathcal{TG}(\psi, b)$: a Truncated Gumbel distribution with location $\psi$ and truncated value $b$ has PDF and CDF at state $g < b$: $f_{\psi+c}(g)/F_{\psi+c}(b), F_{\psi+c}(g)/F_{\psi+c}(b)$.*

### 2.1 GUMBEL PROCESS

Now we are ready to introduce the Gumbel process.

**Definition 3** (Gumbel process (Malmberg, 2013)). *Let $P(S)$ be a sigma-finite measure on sample space $\Omega$, $S \subseteq \Omega$ is a measurable subset. Let $\phi_P(\cdot)$ denote the potential function of $P$ such that $\phi_P(S) = \log P(S) + \log Z_P$. Then $\mathcal{G}_P = \{G_P(S)|S \subseteq \Omega\}$ is a Gumbel process induced from P, if:*

- *(marginal distributions) $G_P(S) \sim \mathcal{G}(\phi_P(S))$.*
- *(independence of disjoint sets) $G_P(S) \perp G_P(S^c)$.*
- *(consistency constraints) for measurable $S_1, S_2 \subseteq \Omega$, then $G_P(S_1 \cup S_2) = \max(G_P(S_1), G_P(S_2))$.*

The Gumbel process is useful in sampling since $\arg\max_{x \in \Omega} G_P(x) \sim P$ (Malmberg, 2013). Therefore, we can draw a sample from $P$ by constructing a Gumbel process for distribution $P$, and then finding the maximum one with some optimization techniques.

In the sequel, we will use $P$ to denote the target distribution, and we call $G_P(S)$ the *Gumbel value* of subset $S$. According to (Malmberg, 2013), Defn. 3 is associated with a natural *bottom-up* construction: for any $x \in \Omega$, we first perturb it with an independent Gumbel noise, i.e., $g(x) \sim \mathcal{G}(0)$. After that we simply set $G_P(x) = g(x) + \phi_P(dx)$ and compute $G_P(S) = \max_{x \in S} G_P(x)$ for all $S \subseteq \Omega$ according to the *consistency constraints*. However, when $\Omega$ is infinite, such a bottom-up construction is infeasible.

**Top-down construction**: (Maddison et al., 2014) presents a top-down construction, which partitions the state space into regions and resolves the problem caused by infinite spaces by considering each region as a whole. Formally, the top-down procedure constructs a top-down tree, $tree(P)$, with each

node corresponding to a subset of $\Omega$. $tree(P)$ is rooted in $\Omega$. Let $par(S)$ denote the parent of subset $S$. For each $S \in tree(P)$, its children is a disjoint partition of $S$, that is, $\cup_{S':par(S')=S}S' = S$.

The top-down construction computes Gumbel values for subsets in the order from the top to the bottom of $tree(P)$. Formally, according to the *consistency constraints* and *marginal distributions*, we compute $G_P(S) \sim \mathcal{TG}(\phi_P(S), L(S))$ where $L(S) := G_P(par(S))$. In the algorithmic view of point, the top-down construction maintains a collection of subsets of $\Omega$. Initially, the collection contains only $\Omega$. At each round, the algorithm selects an element $S$ from the collection and computes $G_P(S)$. After that it divides $S$ into subsets $S_1, S_2$ and adds them into the collection.

## 2.2 $A^*$ SAMPLING

Obviously, if $\phi_P(S)$ is hard to compute, the top-down construction for $P$ is computationally intractable. (Maddison et al., 2014) solves this problem by utilizing the linearity of Gumbel distribution. More specifically, given a distribution $Q$, if $G_Q(x)$ induces a Gumbel process of $Q$, then $G_P(x) := G_Q(x) + \phi_P(x) - \phi_Q(x)$ induces a Gumbel process of distribution $P$. Based on this insight, (Maddison et al., 2014) proposes the $A^*$ sampling, which relies on a tractable proposal distribution $Q$. Furthermore, since $G_Q(S) \perp \arg\max_{x \in S} G_Q(x)$ (Maddison et al., 2014), $A^*$ sampling executes the top-down construction for $Q$, and for each subset, $A^*$ sampling computes $\tilde{G}_P(S) = G_Q(S) + \phi_P(x_Q(S)) - \phi_Q(x_Q(S))$ where $x_Q(S) \sim Q(\cdot|S)$.[1] Suppose at some time point that $A^*$ sampling has processed $n$ nodes in $tree(Q)$, denoted by $Done_Q(n)$. It can be shown that there are $n + 1$ nodes in the *to be processed* collection, denoted by $Collect_Q(n)$. As introduced above, for each $A \in Done_Q(n)$, we have a pair $(x_Q(A), \tilde{G}_P(A))$, and each $S \in Collect_Q(n)$ is associated with a truncated Gumbel distribution $\mathcal{TG}(\phi_Q(S), L(S))$.

The subset selection and termination in $A^*$ sampling rely on a bound function $B : 2^\Omega \to \mathbb{R}$ such that $B(S) \geq \sup_{x \in S} \phi_P(x) - \phi_Q(x)$. Let $D(S) := L(S) + B(S)$. If for some $n$, $\max_{S \in Done(n)} \tilde{G}_P(S) \geq \max_{S \in Collect(n)} D(S)$, $A^*$ terminates and outputs the element with maximum value among the processed nodes. At round $n$, $A^*$ sampling selects the node $S$ with maximum value of $D(S)$ from $Collect(n)$.

## 3 CHALLENGES OF $A^*$ SAMPLING

There are two challenges in $A^*$ sampling. The first one is about the function $D$ on which $A^*$ sampling relies. Computing this function for every $S$ can be intractable since it can be a non-convex optimization. If we simply remove the (possibly intractable) bound function or use a very loose bound, $A^*$ sampling will degenerate to an algorithm which is not efficiency (Maddison et al., 2014). We name the degenerated algorithm as $A^*$ sampling without a bound ( See Appendix D for details.).

The second challenge is that selecting the subset with maximum $D(S)$ is not always a good choice. This is because $D(S)$ is just an upper bound of $G_P(S)$ and it is possible that $G_P(S)$ is relatively small with high probability while $D(S)$ is very large. We now present a simple counter example for $A^*$ sampling to intuitively explain the reason. In this example, $\Omega = (-10.0, +10.0)$, the target is a mixture distribution: $P(x) \propto (1.0 - 10^{-5})\mathcal{N}(x; -5.0, 1.0) + \mathbb{1}[|x| \leq 0.5 * 10^{-405}]10^{400}$ and $Q(x) \propto \mathcal{N}(x; 5.0, 1.0)$. The log likelihoods of $P$ and $Q$ are shown in Fig. 1(a). We first empirically evaluate $A^*$ sampling on this example. Fig. 1(b) shows the selected rounds on the optimal subsets and Fig. 1(c) shows the maximum Gumbel value found by $A^*$ sampling. Results are averaged over 100 runs. We can see that $A^*$ sampling has a poor performance on this example. In this case, $D(S)$ is large if $S$ covers points near $x = 0$. So $A^*$ sampling will allocate lots of computational resources into such intervals, however, $G_P(S)$ being high for such $S$ is with probability only about 0.00001.

## 4 $A^*$ SAMPLING WITH PROBABILITY MATCHING

We now present how to use PM to improve $A^*$ sampling by addressing the above challenges. We first present an intuitive example in Section 4.1. Then, we present a practical PM algorithm based on Monte-Carlo estimators of $G_P(S)$ in Section 4.2 and an efficient implementation with well-designed balanced trees in Section 4.3.

---

[1]In this paper, we use $P(\cdot|S)$ to denote the distribution $P$ conditioned on state space $S$.

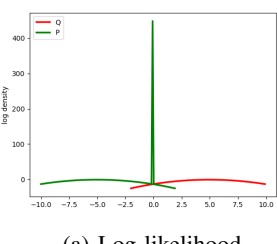
(a) Log-likelihood

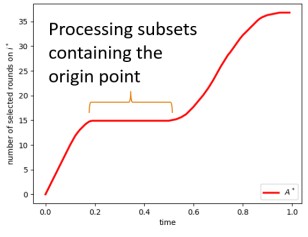
(b) Selected rounds on intervals containing the optimal point

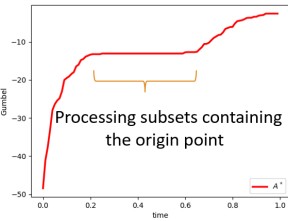
(c) Maximum Gumbel value

Figure 1: The counter example with (a) the log likelihood of $P$ and $Q$; (b) the selected number on the intervals containing the optimal point; (c) the maximum Gumbel value found by $A^*$ sampling.

### 4.1 PROBABILITY MATCHING AND AN EXAMPLE WITH TWO SUBSETS

In general, when making a choice among a set of options, PM selects an option randomly according to its probability of being the optimal. More specifically, in our problem, the optimal option is the subset with the maximum Gumbel value. Formally, by definition, the maximum Gumbel value within region $S$ is a random variable $G_P(S) = \max_{x \in S} \mathcal{TG}(\phi_Q(dx), L(S)) + \phi_P(x) - \phi_Q(x)$. Suppose the state space $\Omega$ is partitioned into $\{S_1, \cdots, S_K\}$. PM selects a subset according to the probability:

$$p\left(i = \arg\max_{k \in [K]} G_P(S_k)\right). \qquad (1)$$

Intuitively, PM has an excellent performance since it allocates computational resources into the options which are likely to have large outcomes. To provide more intuition into why PM suits $A^*$ sampling, we analyze a simple but nontrivial case in which we divide $\Omega$ into two sets.

In order to get a clean theoretical result, we additionally assume $A^*$ sampling does not further split a subset after processing it. We focus on the stochastic regret (Guha & Munagala, 2014) which is the expected number of selections on the suboptimal subset. Formally, suppose $\Omega$ is partitioned into $S_1, S_2$. Let $i^* =$

---

**Algorithm 1** Probability matching with Monte-Carlo estimators.

1: Input: the target distribution $P$, proposal distribution $Q$, state space $\Omega$, time horizon $T$.
2: Output: $x$: a sample from $P$.
3: $maxgumbel = -\infty, x = None$.
4: $Collect = \{\Omega\}, L(\Omega) = \infty, \forall S \subseteq \Omega, t = 1$.
5: **while** $t \leq T$ **do**
6:    $t = t + 1$.
7:    Select $S^*$ according to Eq. (4).
8:    Split $S^*$ into disjoint sets $S_1, S_2$.
9:    $L(S_1) = L(S_2) = G_Q(S^*)$.
10:    **for** $S \in S_1, S_2$ **do**
11:      $x(S) \sim Q(\cdot|S)$.
12:      $G(S) \sim \mathcal{TG}(\phi_Q(S), L(S))$.
13:      $\tilde{G}(S) = G(S) + \phi_P(x(S)) - \phi_Q(x(S))$.

14:      **if** $maxgumbel < \tilde{G}(S)$ **then**
15:        $maxgumbel = \tilde{G}_m(S), x = x(S)$.
16:      **end if**
17:    **end for**
18:    Compute Monte-Carlo estimators for $S_1, S_2$ and update balanced trees.
19:    $Collect.insert(S_1), Collect.insert(S_2)$.
20: **end while**

---

$\arg\max_{i \in \{1,2\}} G_P(S_i)$ which is a random variable. Consider an algorithm $\mathcal{A}$ which selects a subset $S_{i_{\mathcal{A},t}}$ at time step $t$. The stochastic regret of $\mathcal{A}$ at time $T$ is: $R_{\mathcal{A}}(T) = \mathbb{E}\left(\sum_{t=1}^T \mathbb{1}[i_{\mathcal{A},t} \neq i^*]\right)$.

Intuitively, the smaller $R_{\mathcal{A}}$ is, the better $\mathcal{A}$ is, since $\mathcal{A}$ won't waste many computational resources on the suboptimal subset. Moreover, we can prove that PM is optimal within a constant gap in terms of the stochastic regret:

**Lemma 1.** *Let $opt(T)$ denote the algorithm which minimizes the stochastic regret. Then :*

$$R_{PM}(T) \leq 2R_{opt(T)}(T), \ \forall T$$

*where $R_{PM}(T)$ is the stochastic regret of PM.*

The proof of Lemma 1 is adapted from the proof in (Guha & Munagala, 2014) for Bayesian bandits with two arms, we defer the details in Appendix A.

### 4.2 PROBABILITY MATCHING WITH A MONTE-CARLO ESTIMATOR

Unfortunately, drawing samples from the probability in Eq. (1) is intractable when $G_P(S)$ is complex. So in this section, we present an efficient PM algorithm based on a Monte-Carlo estimator of $G_P(S)$.

Consider a random variable $Y = \phi_P(x) - \phi_Q(x), x \sim Q(\cdot|S)$ whose expectation is a constant plussing the KL-divergence between $Q$ and $P$ conditioned on the subset $S$. We can equally characterize $G_P(S)$ as

$$\max_y \mathcal{TG}(\log(Q(S) \cdot p(Y = y)), L(S)) + y. \tag{2}$$

We present the proof of Eq. (2) in Appendix B. Eq. (2) suggests that we can get a Monte-Carlo estimator of $G_P(S)$ by estimating $Y$. More specifically, let $Y_1, \cdots, Y_m$ be a sequence of random variables and $w_1, \cdots, w_m$ be the corresponding weights such that $\sum_{i=1}^{m} w_i = 1, w_i > 0$. Suppose the random variable $\mathcal{Y}_m : p(\mathcal{Y}_m = Y_i) = w_i$ is an unbiased estimator of $Y$, then we can estimate $G_P(S)$ by:

$$\hat{G}_P(S) = \max_{i \in [m]} \mathcal{TG}(\log(w_i Q(S)), L(S)) + Y_i = \max_{i \in [m]} \mathcal{TG}(\log(w_i Q(S)) + Y_i, L(S) + Y_i) \tag{3}$$

The second equality holds due to the linearity of the truncated Gumbel distribution (Maddison et al., 2014). According to Eq. (3), we can estimate $G_P(S)$ with existing Monte-Carlo estimators of $Y$, such as adaptive importance sampling (Gilks & Wild, 1992).

The corresponding PM with Monte-Carlo estimators is to draw samples from

$$p\left(\hat{i} = \arg\max_{j \in [n]} \hat{G}_P(S_j)\right). \tag{4}$$

What remains is how to sample from the probability in Eq. (4) efficiently. The most popular execution of Eq. (4) is as in (Chapelle & Li, 2011): we draw $y_i \sim \hat{G}_P(S_i)$, and take $\hat{i} = \arg\max_i y_i$, then it can be shown that $\hat{i}$ is a sample from the probability in Eq. (4).

However, a direct implementation of the above ideas requires time complexity $O(m)$ since we need to draw samples from $m$ truncated Gumbel distributions, where $m = \sum_{i \in [n]} m_i$ is the number of particles in total and $m_i$ is the number of particles in $S_i$. So our selection algorithm executing $m$ rounds would require running time $O(m^2)$. It is relatively slow comparing with the $O(m \log m)$ time complexity for $A^*$ sampling (Maddison et al., 2014).

### 4.3 AN EFFICIENT IMPLEMENTATION BY BALANCED TREES

We now present a novel algorithm that only requires $O(\log m)$ running time to sample from the distribution in Eq. (4). Our algorithm is based on the properties of the truncated Gumbel distribution and under the help of well-designed balanced trees.

We first decompose sampling from the distribution in Eq. (4) into two steps which can be done efficiently. The decomposition is an immediate inference of Eq. (4):

$$p\left(\hat{i} = \arg\max_{j \in [n]} \hat{G}_P(S_j)\right) = \int_x p(x = \max_{j \in [n]} \hat{G}_P(S)) p(\hat{i} = \arg\max_{j \in [n]} \hat{G}_P(S)|x = \max_{j \in [n]} \hat{G}_P(S)) dx.$$

Thus, sampling from the distribution in Eq. (4) equals to the following two sampling problems:

$$x \sim \max_{i \in [n]} x_i, x_i \sim \hat{G}_P(S_i), \quad \hat{i} \sim p(i = \arg\max_{j \in [n]} x_j \sim \hat{G}_P(S_i)|x = \max x_j)$$

Recall that $\hat{G}_P(S)$ is the maximum one among a set of truncated Gumbels. Thus, the above two sampling problems are essentially sampling the maxima and the argument of the maxima among a set of truncated Gumbels. So our target can be converted into the following problem:

**Problem 1.** *Given a set of truncated Gumbel variables $\{v_i\}_{i=1}^m$ with parameters $(a_i, b_i)$, i.e., $v_i \sim \mathcal{TG}(a_i, b_i)$. We define two sampling problems:*

$$v = \max_{i \in [m]} v_i \tag{5}$$

$$\hat{i} \sim p(i = \arg\max_{j \in [m]} v_j | v = \max_{j \in [m]} v_j) \tag{6}$$

We use the *inverse transform sampling* (Devroye, 1986) to sample $v$ in Eq. (5). In inverse transform sampling, for a random variable $X$ with CDF $U_X(x)$, we first draw a sample $s \sim uniform(0, 1)$, and then compute $x$ such that $U_X(x) = s$, it can be shown that $x \sim X$. Thus, let $U(g)$ denote the CDF of $v$, we only need an algorithm to compute $g$ such that $U(g) = s, s \in (0, 1)$. We now show how to compute such $g$ efficiently with balanced trees.

For notational clarity, let $U_{a,b}(g)$ denote the CDF of a truncated Gumbel distribution, $\mathcal{TG}(a, b)$. According to Defn. 2, we have $U_{a,b}(g) = \frac{\exp(-\exp(-\min(g,b)+a))}{\exp(-\exp(-b+a))}$. Recall $v = \max_i v_i$, then $v$ has CDF: $U(g) = \prod_i U_{a_i,b_i}(g) = \prod_i \frac{\exp(-\exp(-\min(g,b_i)+a_i))}{\exp(-\exp(-b_i+a_i))}$. Take logarithm on both sides, we get $\log U(g) = \sum_{i \in [m]} (-\exp(-\min(g, b_i) + a_i) + \exp(-b_i + a_i))$.

Without loss of generality, we sort $b_i$'s in a non-decreasing order, that is, $b_i \leq b_{i+1}$. Since $U(g)$ is a monotonically increasing function, for $g \in (b_i, b_{i+1}]$, we have:

$$\log U(g) = -\sum_{j>i} \exp(-g + a_j) + \sum_{j>i} \exp(-b_j + a_j) = -\exp(-g) \sum_{j>i} \exp(a_j) + \sum_{j>i} \exp(-b_j + a_j)$$

Thus, given $U(g)$ and suppose $g \in (b_i, b_{i+1}]$, we can compute $g$ by:

$$g = -\log \frac{\sum_{j>i} \exp(a_j - b_j) - \log U(g)}{\sum_{j>i} \exp(a_j)} \tag{7}$$

Thus, when we get $s \sim uniform(0, 1)$, we need to find $i$ such that $U(b_i) \leq s \leq U(b_{i+1})$, and then solve $g$ according to Eq. (7) and inverse sampling . Both of above two steps can be done efficiently via a balanced tree.

Suppose we have a balanced tree such that each node in the tree corresponds to an index $i \in [m]$, and the key of the balanced tree is $b_i$, that is, for all $j$ in the right subtree of node $i$, we have $b_j \geq b_i$ and for all $j$ in the left subtree, we have $b_j \leq b_i$. Suppose that from the balanced tree, we can query in $O(1)$ time at each node $i$ for terms: (1) $\exp(-b_i) \sum_{j>i} \exp(a_j)$; (2) $\exp(a_i) \sum_{j>i} \exp(-b_j)$ and (3) $\sum_{j>i} \exp(a_j - b_j)$. We can query these terms efficiently in a balanced tree because they all are summations over an interval. And according to Defn. 2, we know that $\log U(b_i) = \sum_{j>i} (\exp(-b_j + a_i) - \exp(a_j - b_i))$, we can check out whether $\log s < \log U(b_i)$ in $O(1)$. Therefore, we can find the index $i$ such that $U(b_i) \leq s \leq U(b_{i+1})$ in running time $O(\log m)$. After that, we can compute $U(g) = s$ via Eq. (7) in running time $O(1)$.

Now we turn to sample $\hat{i}$ in Eq. (6). Without loss of generality, suppose $g \in (b_i, b_{i+1}]$. Obviously, for $j < i$, $p(j = \arg\max_{j'} v_{j'} | g = \max_{j'} v_{j'}) = 0$. For $j \geq i$, by Defn. 2 and with simple calculations, we have:

$$p\left(j = \arg\max_{j'} v_{j'} | g\right) \propto \frac{dU_{a_j,b_j}(g)}{dg} \prod_{j' \neq j, j' > i} U_{a_{j'},b_{j'}}(g)$$

$$= \exp(-g + a_j) \frac{\exp(-\exp(-g + a_j))}{\exp(-\exp(-b_j + a_j))} \prod_{j' \neq j, j' > i} U_{a_{j'},b_{j'}}(g) = \exp(a_j) \exp(-g) \prod_{j' > i} U_{a_{j'},b_{j'}}(g) \propto \exp(a_j) \tag{8}$$

According to Eq. (8), we can sample $\hat{i}$ in $O(\log m)$ running time with a balanced tree from which we can query $\sum_{j>i} \exp(a_j)$ efficiently. Putting the previous results together, we get the algorithm as outlined in Alg. 1.

## 5 EXPERIMENTS

In this section, we present our empirical results. We first check the correctness on a simple toy experiment and the counter example in Section 3. After that, we evaluate the efficiency of Alg. 1 on two Bayesian posterior inference tasks, results show that our algorithm outperforms vanilla $A^*$ sampling significantly.

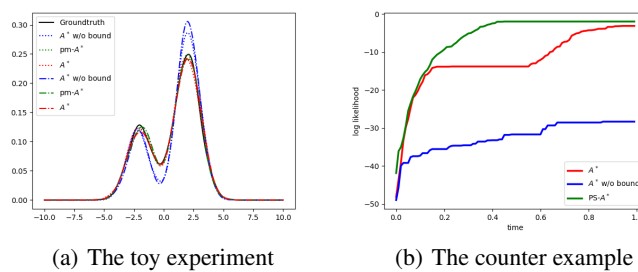

(a) The toy experiment  (b) The counter example

Figure 2: Experiment results with (a) the toy experiment; (b) the counter example.

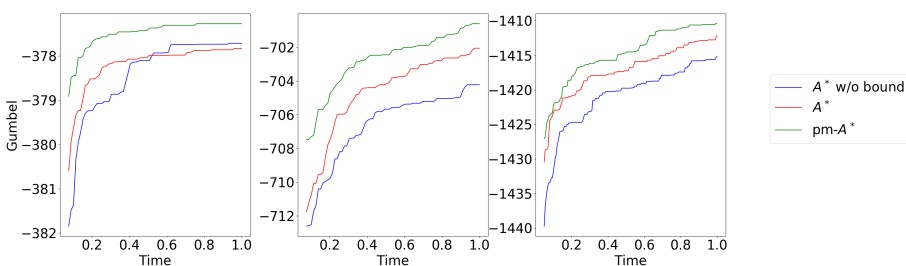

Figure 3: Experiment results on the clutter problem with 5 dimensions on the left, 15 dimensions in the middle and 20 dimensions on the right.

## 5.1 CORRECTNESS OF ALG. 1

We first verify the correctness of Alg. 1 on a greenhouse experiment. We consider sampling from a one-dimensional Gaussian mixture with potential function $\phi_P(x) = -\log(\mathcal{N}(x; -2.0, 1.0) + 2\mathcal{N}(x; 2.0, 1.0))$, which is a multi-mode target distribution. We set $Q = \mathcal{N}(0, 2)$. We present our result in Fig. 2(a) which shows the ground truth and the empirical sample distributions of Alg. 1 and baselines. From Fig. 2(a), Alg. 1 has a similar performance to $A^*$ sampling and outperforms the $A^*$ sampling without a bound function.

## 5.2 THE COUNTER-EXAMPLE

We empirically compare the performance of algorithms on the example in Section 3. The result is shown in Fig. 2(b). We can see that PM-$A^*$ outperforms baselines significantly since our algorithm wastes less resources on suboptimal subsets.

## 5.3 BAYESIAN POSTERIOR INFERENCE

In this section, we evaluate our algorithm on two Bayesian posterior tasks: the clutter problem and the Bayesian logistic regression. More specifically, we focus on sampling tasks of formulations $P(x) \propto p(x) \prod_{i=1}^n p(y_i|x)$ where $x$ is the variable we are going to sample, $p(\cdot)$ is the prior distribution over $x$, and $\{y_i\}_{i=1}^n$ are observations, and $p(y_i|x)$ is the likelihood function. We simply set $Q(x) := p(x)$ in both $A^*$ sampling and Alg. 1. For vanilla $A^*$ sampling, we exploit the standard stochastic gradient descent (SGD) algorithm to calculate the bound function, $\max_{x \in S} \sum_i \log p(y_i|x)$. For the Monte-Carlo estimator in Alg. 1, we exploit the importance sampling over the trajectory of the same SGD algorithm as in the vanilla $A^*$ sampling [2].

## 5.3.1 EVALUATION ON THE CLUTTER PROBLEM

We now evaluate Alg. 1 on the Clutter problem proposed by (Minka, 2001). Clutter problem aims to inference the mean of an isotropic Gaussian with some data points are outliers. Consider a mixture

---

[2]We first tune the parameters of SGD for vanilla $A^*$ sampling, and then apply them to Alg. 1 without further tuning.

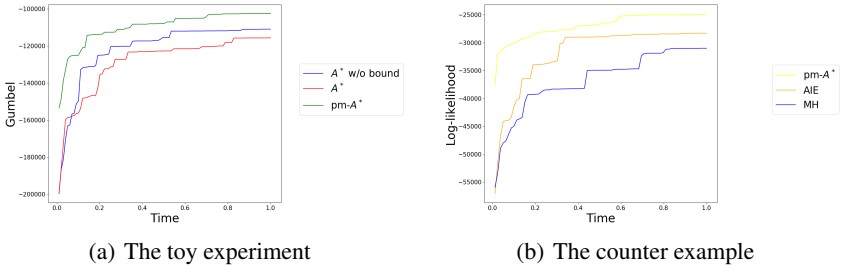

(a) The toy experiment

(b) The counter example

Figure 4: Experiments on logistic regression with (a) averaged Gumbel values; (b) averaged log-likelihoods.

distribution: $p(y|x) = (1-w)\mathcal{N}(y;x,I) + w\mathcal{N}(y;0,\sigma_1 I)$, $p(x) = \mathcal{N}(x;0,\sigma_2 I)$, where $w$ is the ratio of outliers which is a known parameter, and $\mathcal{N}(\mu,\Sigma)$ represents Gaussian distribution. Our goal is to inference $x$ given data $\{y_i\}_{i=1}^n$. We do experiments on dimensions varying in $5, 15, 20$, $n = 20$. We compare the Gumbel of these algorithms. We run 100 times and present the averaged results in Fig. 3. We can see that Alg. 1 outperforms $A^*$ sampling constantly.

### 5.3.2 Evaluation on Bayesian Logistic Regression

Our last experiment is on Bayesian Logistic Regression. Given a dataset $\{x_i\}_{i=1}^n$ associated with label $\{y_i\}_{i=1}^n$ where $y_i \in \{0,1\}$. We follow the setting in (Gershman et al., 2012) and define the Bayesian logistic regression: $p(\alpha) = Gamma(\alpha; a, b)$, $p(w_k) = \mathcal{N}(w_k; 0, \alpha^{-1})$, $p(y_i = 1; x_i, w) = sigmoid(w^T x_i)$. In this model, $\{w, \alpha\}$ are the hidden variables, where $w$ denotes the regression coefficients and $\alpha$ is a precision parameter. We set $a = b = 1$. We do experiments on 13 binary classification datasets proposed by (Mika et al., 1999). The number of features of these data sets are in range from 2 to 60, and the number of points ranges from 24 to 7400 (See Appendix C for more statistics). We present our results in Fig. 4(a) where all results are averaged over 100 runs. Fig. 4(a) presents the summation of the maximum likelihood found by each algorithm on these datasets over time. From Fig. 4(a), we can see that PM-$A^*$ outperforms all baselines.

Furthermore, we compare our algorithm with standard Matropolis-Hastings algorithm (MH) and adaptive inference with exploration (AIE) (Rainforth et al., 2018) which also attempts to bridge the gap between sampling problems and decision-making techniques. For MH, the initial points are sampled from the prior. To make the comparison fair, we also evaluate Alg. 1 and AIE with the prior as the Monte-Carlo estimator instead of gradient-based methods. We compare the likelihoods in Fig. 4(b). We can see that Alg. 1 outperforms AIE even if they use the same Monte-Carlo estimator. This is AIE attempts to use UCB-like algorithm to make decisions, but UCB works only for those models in which concentration bounds hold which is not always valid in sampling problems.

## 6 Conclusion and future work

In this work, we focus on improving the subset selection procedure in $A^*$ sampling with PM. We proved that in the special case of two subsets, PM is optimal within a constant gap in terms of the stochastic regret. Moreover, we proposed a practical algorithm based on Monte-Carlo estimators and well-designed balanced trees. Empirical results show that our methods saves a significantly amount of computational resources on suboptimal regions compared with $A^*$ sampling.

There exists several challenges in future work. The first one is on the analysis of PM. Though we proved PM is efficient in the case of two subsets, it is very challenging to prove the efficiency in general. The second one is that the performance of Alg. 1 relies on the accuracy of the Monte-Carlo estimator. However, it is time-consuming to compute an accurate Monte-Carlo estimator. So it is important to balance the accuracy of the Monte-Carlo estimator and the performance of PM. We hope our work is a starting point to address these problems.

ACKNOWLEDGMENTS

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

## A   PROBABILITY MATCHING FOR TWO SUBSETS

We present the proof of Lemma 1 in this section. This proof is adapted from the proof of Thompson sampling for Bayesian bandits with two arms in (Guha & Munagala, 2014).

We first make the simplified $A^*$ sampling for two subsets clearer. Recall $\Omega$ is divided into two subsets $S_1, S_2$. For notational convenience, let $k_{t,i}$ denote the number of selected rounds of $S_i$ up to time step $t$. Since we do not split $S_1, S_2$, if we have processed $S_i$ for $t$ rounds, we will have $t$ observations $\{(x_{i,j}, u_{i,j})\}_{j=1}^t$ such that $x_{i,1} \sim Q(\cdot|S_i), u_{i,1} \sim \mathcal{G}(\phi_Q(S_i))$ and $x_{i,j} \sim Q(\cdot|S_i \backslash X_{i,j-1}), u_{i,j} \sim \mathcal{TG}(\phi_Q(S_i \backslash X_{i,j-1}), u_{i,j-1})$ for $j > 1$ where $X_{i,j} = \{x_{i,j'}\}_{j'=1}^j$. Let $V_{i,t} = \{(x_{i,j}, u_{i,j})\}_{j=1}^{k_{t,i}}$ denote the observations from subset $S_i$ until time $t$ and $V_t = \{V_{1,t}, V_{2,t}\}$ denote the collection of observations from both subsets. Obviously, a subset selection algorithm $\mathcal{A}$ selects a subset according to previous observations, that is, we can represent the subset selected by $\mathcal{A}$ at time $t$, $i_{\mathcal{A},t+1}$, as $i_{\mathcal{A},t+1} := i_{\mathcal{A}}(V_t)$. Now we present $A^*$ sampling for two subsets with $\mathcal{A}$ as the subset selection algorithm in Alg. 2.

---

**Algorithm 2** $A^*$ sampling for two subsets.

---

1: Input: the target distribution $P$, proposal distribution $Q$, state space $\Omega = S_1 \cup S_2$, a subset selection algorithm $\mathcal{A}$, time horizon $T$.
2: Output: $x$: a sample from $P$.
3: $maxgumbel = -\infty, x = None$.
4: $u_{1,0} = u_{2,0} = \infty, t = 0$.
5: **while** $t \leq T$ **do**
6:    $t = t + 1$.
7:    $i = i_{\mathcal{A}}(V_{t-1})$.
8:    $x' \sim Q(\cdot|S_i \backslash X_{i,k_{t-1,i}})$.
9:    $u_{i,t} \sim \mathcal{TG}(\phi_Q(S_i \backslash X_{i,k_{t-1,i}}), u_{i,t-1})$.
10:   $u' = u_{i,t} + \phi_P(x(S)) - \phi_Q(x(S))$.
11:   **if** $maxgumbel < u'$ **then**
12:     $maxgumbel = u', x = x'$.
13:   **end if**
14: **end while**

---

Recall $opt(T)$ denote the the algorithm which minimizes the stochastic regret up to time horizon $T$. We now explicitly define the optimal stochastic regret $R_{opt(T)}(T)$ with time horizon $T$ via dynamic programming. Let $q_i(V)$ denote the probability of $S_i$ being the optimal subset given observations $V$. For convenience, let $R_{\mathcal{A}}(T|V)$ be the stochastic regret of algorithm $\mathcal{A}$ given observations $V$. Then:

**Definition 4.** *We can define $R_{opt(T)}(T|V)$ as follows:*

$$R_{opt(1)}(1|V) = \min_{i \in \{1,2\}} (1 - q_i(V)).$$

$$\forall T > 1, R_{opt(T)}(T|V) = \min_{i \in \{1,2\}} \left( 1 - q_i(V) + \int_{x_i, u_i} R_{opt(T-1)}(T-1|V \cup (x,u)) p(x,u|V) \right)$$

*where $(x_i, u_i)$ is the new observation from subset $S_i$.*

We now present the formal definition and some properties of $q_i(V)$ which are useful in the sequel:

**Corollary 1.** *Let $V = \{V_1 := \{(x_{1,j}, u_{1,j})\}_{j=1}^{k_1}, V_2 := \{(x_{2,j}, u_{2,j})\}_{j=1}^{k_2}\}$ denote a set of observations. Let $S_{i,\backslash V} = S_i \backslash \cup \{x_{i,j}\}_{j=1}^{k_i}$. Then let $i' = \{1,2\}\backslash i$ and $m(x) = \phi_P(x) - \phi_Q(x)$, we have:*

$$q_i(V) = p \left( \max_{x \in S_{i,\backslash V}} \mathcal{TG}(\phi_Q(dx), u_{i,k_i}) + m(x) > \max_{x \in S_{i',\backslash V}} \mathcal{TG}(\phi_Q(dx), u_{i',k_{i'}}) + m(x) \right)$$

*Suppose we process $S_i$ for the $k_i + 1$ time, and then receive observation $(x_i', u_i')$, we have:*

$$q_i(V) \geq \mathbb{E}_{x_i', u_i'} [q_i(V \cup (x_i', u_i'))] \tag{9}$$

*Proof.* Let $\mathcal{E}$ denote the event of the $(k_i + 1)$-th observation from subset $i$ is with the maximum Gumbel value within $S_i$, that is, $u_i' + m(x_i') \geq \max_{S_{i,\backslash(V \cup x_i')}} \mathcal{TG}(\phi_Q(dx), u_i') + m(x)$. Then we have $q_i(V \cup (x_i', u_i')|\bar{\mathcal{E}}) = q_i(V|\bar{\mathcal{E}})$; otherwise, $q_i$ will decrease, that is, $q_i(V \cup (x_i', u_i')|\mathcal{E}) \leq q_i(V|\mathcal{E})$. Overall, we have

$$\mathbb{E}_{x_i', u_i'}[q_i(V \cup (x_i', u_i'))] = p(\mathcal{E})q_i(V \cup (x_i', u_i')|\mathcal{E}) + p(\bar{\mathcal{E}})q_i(V \cup (x_i', u_i')|\bar{\mathcal{E}} \leq q_i(V)$$

$\square$

For convenience, let $R_\mathcal{A}(T|V) = \sum_i p(i_{\mathcal{A},1} = i)R_\mathcal{A}(i, T|V)$ where $R_\mathcal{A}(i, T|V)$ denote the stochastic regret of $\mathcal{A}$ if we select subset $i$ at the first round. The proof of Lemma 1 relies on the following lemma:

**Lemma 2.** *Let $P$ denote the target distribution and $Q$ denote the proposal. If an algorithm $\mathcal{A}$ satisfies that for any observations $V$ and time horizon $T$:*

$$R_\mathcal{A}(T|V) \leq R_\mathcal{A}(i, T|V) + c(1 - q_i(V)) \tag{10}$$

*where $c$ is a constant. Then we have $R_\mathcal{A}(T|V) \leq (c+1)R_{opt(T)}(T|V)$ for all $T$ and $V$.*

*Proof.* For $T = 1$, it is obvious that Eq. (10) holds. We use mathematical induction to prove the case of $T > 1$. According to definition and with straight-forward calculations, we have:

$$
\begin{aligned}
R_\mathcal{A}(T|V) &\leq R_\mathcal{A}(i, T|V) + c(1 - q_i(V)), & \forall i \\
&= (c+1)(1 - q_i(V)) + \mathbb{E}_{(x,u)}R_\mathcal{A}(i, T-1|V \cup (x, u)), & \forall i \\
&\leq (c+1)(1 - q_i(V) + \mathbb{E}_{(x,u)}R_{opt(T-1)}(i, T-1|V \cup (x, u))), & \forall i \\
&\leq (c+1)\min_i(1 - q_i(V) + \mathbb{E}_{(x,u)}R_{opt(T-1)}(i, T-1|V \cup (x, u))) \\
&= (c+1)R_{opt(T)}(i, T|V)
\end{aligned}
$$

$\square$

According to Lemma 2, we can prove Lemma 1 by proving that probability matching satisfies precondition (10) for $c = 1$. The following Lemma provides an equivalent characterization of Eq. (10) in the context of probability matching.

**Lemma 3.** *For probability matching, we have:*

$$R_{PM}(T) \leq R_{PM}(i, T) + 1 - q_i, \forall i \in \{1, 2\} \iff |R_{PM}(1, T) - R_{PM}(2, T)| \leq 1 \tag{11}$$

*Proof.* Suppose $R_{PM}(T) \leq R_{PM}(1, T) + 1 - q_1$. By definition, we have $R_{PM}(T) = q_1 R_{PM}(1, T) + q_2 R_{PM}(2, T)$. Observing $q_1 + q_2 = 1$, we have $R_{PM}(2, T) - R_{PM}(1, T) \leq 1$. Conversely, if $R_{PM}(1, T) \leq R_{PM}(2, T) + 1$, we have $R_{PM}(T) = q_1 R_{PM}(1, T) + q_2 R_{PM}(2, T) \leq R_{PM}(2, T) + 1 - q_2$. Swapping the roles of 1 and 2, we complete the proof. $\square$

Now, we can complete the proof of Lemma 1.

*Proof.* For $T = 1$, it is obvious that Lemma 3 is true. We use mathematical induction to prove Lemma 3 holds for $T > 1$. For convenience, let $u_1 \sim \mathcal{TG}(\phi_Q(S_1), L(S_1)), x_1 \sim Q(\cdot|S_1), u_2 \sim \mathcal{TG}(\phi_Q(S_2), L(S_2)), x_2 \sim Q(\cdot|S_2)$. We have

$$
\begin{aligned}
R_{PM}(1, T|V) &= 1 - q_1(V) + \mathbb{E}_{x_1, u_1}[R_{PM}(T-1|V \cup (x_1, u_1))] \\
&\leq 1 - q_1(V) + \mathbb{E}_{x_1, u_1}[1 - q_2(V \cup (x_1, u_1)) + R_{PM}(2, T-1|V \cup (x_1, u_1))] \\
&= 1 - q_1(V) + \mathbb{E}_{x_1, u_1}[2q_1(V \cup (x_1, u_1)) + \mathbb{E}_{x_2, u_2}R_{PM}(T-2|V \cup \{(x_1, u_1), (x_2, u_2)\})] \\
&\leq 1 + q_1(V) + \mathbb{E}_{x_1, u_1}[\mathbb{E}_{x_2, u_2}R_{PM}(T-2|V \cup \{(x_1, u_1), (x_2, u_2)\})]
\end{aligned}
$$

The first equation is according to the inductive hypothesis, the next equality follows $q_1(V) + q_2(V) = 1$ and the last inequality is due to Corollary 1. Similarly, we have:

$$
\begin{aligned}
R_{PM}(2, T|V) &= 1 - q_2(V) + \mathbb{E}_{x_2, u_2}[R_{PM}(T-1|V \cup (x_2, u_2))] \\
&= q_1(V) + \mathbb{E}_{x_2, u_2}[q_1(V \cup (x_2, u_2))R_{PM}(1, T-1|V \cup (x_2, u_2)) \\
&\quad + q_2(V \cup (x_2, u_2))R_{PM}(2, T-1|V \cup (x_2, u_2))] \\
&\geq q_1(V) + \mathbb{E}_{x_2, u_2}[R_{PM}(1, T-1|V \cup (x_2, u_2)) - q_2(V \cup (x_2, u_2))] \\
&= q_1(V) + \mathbb{E}_{x_2, u_2}[\mathbb{E}_{x_1, u_1}R_{PM}(T-2|(x_1, u_1), (x_2, u_2))]
\end{aligned}
$$

The first inequality follows the inductive hypothesis in Lemma 3 and $q_1(V) + q_2(V) = 1$. Above inequalities show that $R_{PM}(1, T) - R_{PM}(2, T) \leq 1$. Reversing the roles of $S_1$ and $S_2$, we can show that probability matching satisfies the condition in Lemma 3 which completes the proof. $\square$

## B    THE EQUIVALENT CHARACTERIZATION OF $G_P(S)$

We now prove the equivalent characterization of $G_P(S)$ in Section 4.2. According to definitions, we have:

$$
\begin{aligned}
G_P(S) &= \max_{x \in S} \mathcal{TG}(\phi_Q(dx), L(S)) + \phi_P(dx) - \phi_Q(x) \\
&= \max_y \max_{\phi_P(x) - \phi_Q(x) = y} \mathcal{TG}(\phi_Q(dx), L(S)) + y \\
&= \max_y (y + \mathcal{TG}(\log(Q(S)p(Y = y)), L(S)))
\end{aligned}
$$

the first equation is by definition and the third equation follows the fact that $\max_{x \in S} \mathcal{TG}(\phi_Q(dx), L) = \mathcal{TG}(\log(\int_{x \in S} \exp(\phi_Q(x))dx), L)$.

## C    STATISTICS OF DATASETS PROPOSED BY (MIKA ET AL., 1999)

We present statistics of datasets proposed by (Mika et al., 1999) in Table 1 where *n* denotes the number of data points and *dim* denotes dimension.

|     | Banana | B.Cancer | Diabetes | German | Heart | Image | Ringnorm |
|-----|--------|----------|----------|--------|-------|-------|----------|
| dim | 2      | 9        | 8        | 20     | 5     | 18    | 20       |
| n   | 5300   | 263      | 768      | 1000   | 215   | 2086  | 7400     |
|     | F.Sonar | Splice  | Thyroid  | Titanic | Twonorm | Waveform | |
| dim | 9      | 60       | 5        | 3      | 20    | 21    |          |
| n   | 144    | 2991     | 215      | 24     | 7400  | 5000  |          |

Table 1: Some statistics of datasets proposed by (Mika et al., 1999).

## D    $A^*$ SAMPLING WITHOUT THE BOUND FUNCTION

As presented in Section 3, when the bound $B(S)$ is not available or very loose, the vanilla $A^*$ sampling degenerates to Alg .3 which is not very efficient since it selects subsets simply according to $L(S)$.

---

**Algorithm 3** $A^*$ sampling without a bound.

---

1: Input: the target distribution $P$, proposal distribution $Q$, state space $\Omega$, time horizon $T$.
2: Output: $x$: a sample from $P$.
3: $maxgumbel = -\infty, x = None$.
4: $Collect = \{\Omega\}, L(\Omega) = \infty, \forall S \subseteq \Omega, t = 1$.
5: **while** $t \leq T$ **do**
6:     $t = t + 1$.
7:     Select $S^* \in Collect$ with maximum $L(\Omega)$.
8:     Split $S^*$ into disjoint sets $S_1, S_2$.
9:     $L(S_1) = L(S_2) = G_Q(S^*)$.
10:    **for** $S \in S_1, S_2$ **do**
11:       $x(S) \sim Q(\cdot|S)$.
12:       $G(S) \sim \mathcal{TG}(\phi_Q(S), L(S))$.
13:       $\tilde{G}(S) = G(S) + \phi_P(x(S)) - \phi_Q(x(S))$.
14:       **if** $maxgumbel < \tilde{G}(S)$ **then**
15:          $maxgumbel = \tilde{G}_m(S), x = x(S)$.
16:       **end if**
17:    **end for**
18:    $Collect.insert(S_1), Collect.insert(S_2)$.
19: **end while**

---

