# OpenReview forum: "$A^*$ sampling with probability matching"
_ICLR.cc/2019/Conference_

### Official Review · AnonReviewer3 · 2018-10-20

**Rating:** 3
**Confidence:** 5

**Review:**

This paper proposes an alternative search procedure for A* sampling that, in contrast to the original optimistic search, doesn't rely on (possibly difficult-to-find) bounds for the log-probability function.

The first major issue with this paper is clarity. The preliminary section describing the Gumbel process and A* sampling is very difficult to understand (despite my being quite familiar with A* sampling). The authors use undefined notation frequently throughout the introduction and refer to it in abstract terms. There are also numerous errors -- for example, when describing the "bottom up" approach to generating a Gumbel process, the authors suggest perturbing all points in the input space by independent Gumbel noise (which would result in Gp(S) = infty almost surely when omega is uncountable).

The description of the main contributions in section 4 is equally unclear. This section starts by suggesting that sampling the next node to investigate according to the probability that it contains the maximum is reasonable, and then presents a lemma about regret in a bandit setting where the sampler never splits the subset. This lemma does not apply to the actual sampler proposed in the paper, so it is not clear why it is included. Section 4.2 is also very unclear -- I am not certain how both w and script Y are defined, nor why we need an "unbiased estimator of Y" (a random variable?) when we can simply sample from Y directly. As the definition of w is unclear, the purpose of 4.3 is unclear as well.

The other major issue is more fundamental -- I am not convinced the sampler is correct. The algorithm simply terminates after some finite horizon (rather than having a conclusive proof of termination via branch and bound as in the original A*). There is no proof or argument included in the paper regarding this. Any proposed sampling algorithm must be correct to be acceptable in ICLR.

---

### Official Review · AnonReviewer2 · 2018-11-02
**good paper**

**Rating:** 6
**Confidence:** 2

**Review:**

 It seems a good paper,  containing with interesting ideas.
However, the state-of-the-art discussion about sampling methods is formed by just one sentence. I suggest to add more references about adaptive rejection sampling and MCMC, discussing a bit more their differences, and the differences and connections with A* sampling. Also direct methods should be discussed. For this reasons, I also suggest to add references to related books as Devroye's sampling book  and other, more recent books devoted to describe independent random sampling methods.

---

### Official Review · AnonReviewer1 · 2018-11-02
**Strong contribution to family of A* sampling algorithms, but lacks clarity.**

**Rating:** 5
**Confidence:** 5

**Review:**

Summary: This paper introduces a probability matching approach for optimizing Gumbel processes, i.e. the extension of the Gumbel-Max trick to more general measure spaces. The basic idea is to use a more refined subset selection mechanism as compared to A* Sampling, but at the cost of being able to guarantee an exact sample. Instead, the authors study the algorithm's best Gumbel so far as a function of the time budget.

Quality: This is an interesting and natural idea, and the experiments support the author's claim that it improves over A* Sampling in the regimes that they consider. The claims and proofs look correct to me.

Originality: This idea has not been explored in the literature, and is a sensible one to consider when having an exact sample is not necessary.

Clarity: The comparison to A* Sampling is a bit apples-to-oranges and the paper lacks some clarity overall. In particular, the authors should be much more clear that their termination condition necessarily differs from A* Sampling's. In A* Sampling the termination condition guarantees that the algorithm has found the true optimum, but in PM-A* the authors consider only a fixed budget of time --- terminating whether or not the algorithm has found the true optimum. The difference is a necessary feature, because otherwise I'm fairly sure A* Sampling is optimal (It has to push the upper bound of every subset below the true maximum, so if the term. cond. is not currently satisfied the subset with the current max. upper bound must eventually be visited before termination. Since it is the only subset where that is necessarily true, taking it is the optimal choice.) More comments below.

Significance: I think this is interesting work, but I would recommend the authors consider some examples that clarify where these ideas might impact that problems that the ICLR audience would be interested in.

Overall: The paper's clarity problems get in the way of its contribution. I think it is an interesting direction, but the authors are not clear enough about the correctness of the algorithm (it is not guaranteed to return an exact sample) to recommend acceptance. If this and other issues are fixed, it would be a good paper.

Pros:
- Solid contribution to the literature on Gumbels.
- Beginnings of a regret analysis.
- Uniform performance gains in comparison to A* Sampling (in the regime considered).

Cons:
- Lacks clarity in comparisons to A* Sampling.
- Writing could clarify significance as well.

Specific questions / suggestions:

- One of the important omissions of the paper is the following question. What is the distribution of PM-A*'s output at time T for a fixed T? It won't be the target distribution in general, but it may be close. This applicability of this work will be greatly improved by that analysis.

- In general the figures and experiments need more details. In particular,
(1) In figure 2a, why are there two sets of lines? Different values of T?
(2) You need to report T throughout.
(3) What did you take as Y? This is a critical part of the method, and you should report the estimator used.

- In A* Sampling for two subsets without splitting algorithm,  shouldn't you be removing ALL previous Xi from  be S_i? i.e. S_i \ {Xi_1, Xi_2, ... Xi_{k-1}} or something like that.

---

### Meta-Review · Area_Chair1 · 2018-12-11
**Novel idea for A* sampling, but lacks clarity**

**Confidence:** 5
**Recommendation:** Reject

**Metareview:**

This paper applied probability matching to A* sampling in order to provide an approximate variant without a bound function. It is a novel idea and a good contribution to the A* sampling family. The authors also provided regret analysis for the adoption of PM.

However, as pointed out by R1 and R3, the authors failed to clarify the approximation introduced by the PM and its implication in the output samples. The empirical comparison should also take into account this difference. Further analysis of the bias in the sample distribution would also help clarify the pros and cons of the proposed method.

R3 also raised the concern that the description of the preliminary section and the main contribution in section 4 was not clear.